# Neural Topic Model via Optimal Transport

**He Zhao, Dinh Phung, Viet Huynh, Trung Le, Wray Buntine**
Department of Data Science and Artificial Intelligence, Faculty of Information Technology
Monash University, Australia
`{ethan.zhao,dinh.phung,viet.huynh,trunglm,wray.buntine}@monash.edu`

## Abstract

Recently, Neural Topic Models (NTMs) inspired by variational autoencoders have obtained increasingly research interest due to their promising results on text analysis. However, it is usually hard for existing NTMs to achieve good document representation and coherent/diverse topics at the same time. Moreover, they often degrade their performance severely on short documents. The requirement of reparameterisation could also comprise their training quality and model flexibility. To address these shortcomings, we present a new neural topic model via the theory of optimal transport (OT). Specifically, we propose to learn the topic distribution of a document by directly minimising its OT distance to the document's word distributions. Importantly, the cost matrix of the OT distance models the weights between topics and words, which is constructed by the distances between topics and words in an embedding space. Our proposed model can be trained efficiently with a differentiable loss. Extensive experiments show that our framework significantly outperforms the state-of-the-art NTMs on discovering more coherent and diverse topics and deriving better document representations for both regular and short texts.

## 1 Introduction

As an unsupervised approach, topic modelling has enjoyed great success in automatic text analysis. In general, a topic model aims to discover a set of latent topics from a collection of documents, each of which describes an interpretable semantic concept. Topic models like Latent Dirichlet Allocation (LDA) (Blei et al., 2003) and its hierarchical/Bayesian extensions, e.g., in Blei et al. (2010); Paisley et al. (2015); Gan et al. (2015); Zhou et al. (2016) have achieved impressive performance for document analysis. Recently, the developments of Variational AutoEncoders (VAEs) and Autoencoding Variational Inference (AVI) (Kingma & Welling, 2013; Rezende et al., 2014) have facilitated the proposal of Neural Topic Models (NTMs) such as in Miao et al. (2016); Srivastava & Sutton (2017); Krishnan et al. (2018); Burkhardt & Kramer (2019). Inspired by VAE, many NTMs use an encoder that takes the Bag-of-Words (BoW) representation of a document as input and approximates the posterior distribution of the latent topics. The posterior samples are further input into a decoder to reconstruct the BoW representation. Compared with conventional topic models, NTMs usually enjoy better flexibility and scalability, which are important for the applications on large-scale data.

Despite the promising performance and recent popularity, there are several shortcomings for existing NTMs, which could hinder their usefulness and further extensions. **i)** The training and inference processes of NTMs are typically complex due to the prior and posterior constructions of latent topics. To encourage topic sparsity and smoothness, Dirichlet (Burkhardt & Kramer, 2019) or gamma (Zhang et al., 2018) distributions are usually used as the prior and posterior of topics, but reparameterisation is inapplicable to them, thus, complex sampling schemes or approximations have to be used, which could limit the model flexibility. **ii)** A desideratum of a topic model is to generate better topical representations of documents with more coherent and diverse topics; but for many existing NTMs, it is hard to achieve good document representation and coherent/diverse topics at the same time. This is because the objective of NTMs is to achieve lower reconstruction error, which usually means topics are less coherent and diverse, as observed and analysed in Srivastava & Sutton (2017); Burkhardt & Kramer (2019). **iii)** It is well-known that topic models degrade their performance severely on short documents such as tweets, news headlines and product reviews, as each individual document contains insufficient word co-occurrence information. This issue can be exacerbated for NTMs because of the use of the encoder and decoder networks, which are usually more vulnerable to data sparsity.

To address the above shortcomings for NTMs, we in this paper propose a neural topic model, which is built upon a novel Optimal Transport (OT) framework derived from a new view of topic modelling. For a document, we consider its content to be encoded by two representations: the observed representation, $x$, a distribution over all the words in the vocabulary and the latent representation, $z$, a distribution over all the topics. $x$ can be obtained by normalising a document's word count vector while $z$ needs to be learned by a model. For a document collection, the vocabulary size (i.e., the number of unique words) can be very large but one individual document usually consists of a tiny subset of the words. Therefore, $x$ is a sparse and low-level representation of the semantic information of a document. As the number of topics is much smaller than the vocabulary size, $z$ is the relatively dense and high-level representation of the same content. Therefore, the learning of a topic model can be viewed as the process of learning the distribution $z$ to be as close to the distribution $x$ as possible. Accordingly, it is crucial to investigate how to measure the distance between two distributions with different supports (i.e., words to $x$ and topics to $z$). As optimal transport is a powerful tool for measuring the distance travelled in transporting the mass in one distribution to match another given a specific cost function, and recent development on computational OT (e.g., in Cuturi (2013); Frogner et al. (2015); Seguy et al. (2018); Peyré et al. (2019)) has shown the promising feasibility to efficiently compute OT for large-scale problems, it is natural for us to develop a new NTM based on the minimisation of OT.

Specifically, our model leverages an encoder that outputs topic distribution $z$ of a document by taking its word count vector as input like standard NTMs, but we minimise the OT distance between $x$ and $z$, which are two discrete distributions on the support of words and topics, respectively. Notably, the cost function of the OT distance specifies the weights between topics and words, which we define as the distance in an embedding space. To represent their semantics, all the topics and words are embedded in this space. By leveraging the pretrained word embeddings, the cost function is then a function of topic embeddings, which will be learned jointly with the encoder. With the advanced properties of OT on modelling geometric structures on spaces of probability distributions, our model is able to achieve a better balance between obtaining good document representation and generating coherent/diverse topics. In addition, our model eases the burden of designing complex sampling schemes for the posterior of NTMs. More interestingly, our model is a natural way of incorporating pretrained word embeddings, which have been demonstrated to alleviate the issue of insufficient word co-occurrence information in short texts (Zhao et al., 2017; Dieng et al., 2020). With extensive experiments, our model can be shown to enjoy the state-of-the-art performance in terms of both topic quality and document representations for both regular and short texts.

## 2 BACKGROUND

In this section, we recap the essential background of neural topic models and optimal transport.

### 2.1 NEURAL TOPIC MODELS

Most of existing NTMs can be viewed as the extensions of the framework of VAEs where the latent variables can be interpreted as topics. Suppose the document collection to be analysed has $V$ unique words (i.e., vocabulary size). Each document consists of a word count vector denoted as $x \in \mathbb{N}^V$ and a latent distribution over $K$ topics: $z \in \mathbb{R}^K$. An NTM assumes that $z$ for a document is generated from a prior distribution $p(z)$ and $x$ is generated by the conditional distribution $p_\phi(x|z)$ that is modelled by a decoder $\phi$. The model's goal is to infer the topic distribution given the word counts, i.e., to calculate the posterior $p(z|x)$, which is approximated by the variational distribution $q_\theta(z|x)$ modelled by an encoder $\theta$. Similar to VAEs, the training objective of NTMs is the maximisation of the Evidence Lower BOund (ELBO):

$$\max_{\theta,\phi} \left( \mathbb{E}_{q_\theta(z|x)} \left[ \log p_\phi(x|z) \right] - \mathbb{KL} \left[ q_\theta(z|x) \parallel p(z) \right] \right). \tag{1}$$

The first term above is the expected log-likelihood or reconstruction error. As $x$ is a count-valued vector, it is usually assumed to be generated from the multinomial distribution: $p_\phi(x|z) := \text{Multi}(\phi(z))$, where $\phi(z)$ is a probability vector output from the decoder. Therefore, the expected log-likelihood is proportional to $x^T \log \phi(z)$. The second term is the Kullback–Leibler (KL) divergence that regularises $q_\theta(z|x)$ to be close to its prior $p(z)$. To interpret topics with words, $\phi(z)$ is usually constructed by a single-layer network (Srivastava & Sutton, 2017): $\phi(z) := \text{softmax}(\mathbf{W}z)$, where $\mathbf{W} \in \mathbb{R}^{V \times K}$

indicates the weights between topics and words. Different NTMs may vary in the prior and the posterior of $\boldsymbol{z}$, for example, the model in Miao et al. (2017) applies Gaussian distributions for them and Srivastava & Sutton (2017); Burkhardt & Kramer (2019) show that Dirichlet is a better choice. However, reparameterisation cannot be directly applied to a Dirichlet, so various approximations and sampling schemes have been proposed.

## 2.2 OPTIMAL TRANSPORT

OT distances have been widely used for the comparison of probabilities. Here we limit our discussion to OT for discrete distributions, although it applies for continuous distributions as well. Specifically, let us consider two probability vectors $\boldsymbol{r} \in \Delta^{D_r}$ and $\boldsymbol{c} \in \Delta^{D_c}$, where $\Delta^D$ denotes a $D-1$ simplex. The OT distance[1] between the two probability vectors can be defined as:

$$d_{\mathbf{M}}(\boldsymbol{r}, \boldsymbol{c}) := \min_{\mathbf{P} \in U(\boldsymbol{r}, \boldsymbol{c})} \langle \mathbf{P}, \mathbf{M} \rangle , \tag{2}$$

where $\langle \cdot, \cdot \rangle$ denotes the Frobenius dot-product; $\mathbf{M} \in \mathbb{R}_{\geq 0}^{D_r \times D_c}$ is the cost matrix/function of the transport; $\mathbf{P} \in \mathbb{R}_{>0}^{D_r \times D_c}$ is the transport matrix/plan; $U(\boldsymbol{r}, \boldsymbol{c})$ denotes the transport polytope of $\boldsymbol{r}$ and $\boldsymbol{c}$, which is the polyhedral set of $D_r \times D_c$ matrices: $U(\boldsymbol{r}, \boldsymbol{c}) := \{P \in \mathbb{R}_{>0}^{D_r \times D_c} | P \mathbf{1}_{D_c} = \boldsymbol{r}, P^T \mathbf{1}_{D_r} = \boldsymbol{c}\}$; and $\mathbf{1}_D$ is the $D$ dimensional vector of ones. Intuitively, if we consider two discrete random variables $X \sim \text{Categorical}(\boldsymbol{r})$ and $Y \sim \text{Categorical}(\boldsymbol{c})$, the transport matrix $\mathbf{P}$ is a joint probability of $(X, Y)$, i.e., $\text{p}(X = i, Y = j) = p_{ij}$ and $U(\boldsymbol{r}, \boldsymbol{c})$ is the set of all the joint probabilities. The above optimal transport distance can be computed by finding the optimal transport matrix $\mathbf{P}^*$. It is also noteworthy that the Wasserstein distance can be viewed as a specific case of the OT distances.

As directly optimising Eq. (2) can be time-consuming for large-scale problems, a regularised optimal transport distance with an entropic constraint is introduced in Cuturi (2013), named the Sinkhorn distance:

$$d_{\mathbf{M}, \alpha}(\boldsymbol{r}, \boldsymbol{c}) := \min_{\mathbf{P} \in U_\alpha(\boldsymbol{r}, \boldsymbol{c})} \langle \mathbf{P}, \mathbf{M} \rangle , \tag{3}$$

where $U_\alpha(\boldsymbol{r}, \boldsymbol{c}) := \{\mathbf{P} \in U(\boldsymbol{r}, \boldsymbol{c}) | h(\mathbf{P}) \geq h(\boldsymbol{r}) + h(\boldsymbol{c}) - \alpha\}$, $h(\cdot)$ is the entropy function, and $\alpha \in [0, \infty)$. To compute the Sinkhorn distance, a Lagrange multiplier is introduced for the entropy constraint to minimise Eq. (3), resulting in the Sinkhorn algorithm, widely-used for discrete OT problems.

## 3 PROPOSED MODEL

Now we introduce the details of our proposed model. Specifically, we present each document as a distribution over $V$ words, $\tilde{\boldsymbol{x}} \in \Delta^V$ obtained by normalising $\boldsymbol{x}$: $\tilde{\boldsymbol{x}} := \boldsymbol{x}/S$ where $S := \sum_{v=1}^{V} \boldsymbol{x}$ is the length of a document. Also, each document is associated with a distribution over $K$ topics: $\boldsymbol{z} \in \Delta^K$, each entry of which indicates the proportion of one topic in this document. Like other NTMs, we leverage an encoder to generate $\boldsymbol{z}$ from $\tilde{\boldsymbol{x}}$: $\boldsymbol{z} = \text{softmax}(\theta(\tilde{\boldsymbol{x}}))$. Notably, $\theta$ is implemented with a neural network with dropout layers for adding randomness. As $\tilde{\boldsymbol{x}}$ and $\boldsymbol{z}$ are two distributions with different supports for the same document, to learn the encoder, we propose to minimise the following OT distance to push $\boldsymbol{z}$ towards $\tilde{\boldsymbol{x}}$:

$$\min_{\theta} d_{\mathbf{M}}(\tilde{\boldsymbol{x}}, \boldsymbol{z}) . \tag{4}$$

Here $\mathbf{M} \in \mathbb{R}_{>0}^{V \times K}$ is the cost matrix, where $m_{vk}$ indicates the semantic distance between topic $k$ and word $v$. Therefore, each column of $\mathbf{M}$ captures the importance of the words in the corresponding topic. In addition to the encoder, $\mathbf{M}$ is a variable that needs to be learned in our model. However, learning the cost function is reported to be a non-trivial task (Cuturi & Avis, 2014; Sun et al., 2020). To address this problem, we specify the following construction of $\mathbf{M}$:

$$m_{vk} = 1 - \cos(\boldsymbol{e}_v, \boldsymbol{g}_k) , \tag{5}$$

where $\cos(\cdot, \cdot) \in [-1, 1]$ is the cosine similarity; $\boldsymbol{g}_k \in \mathbb{R}^L$ and $\boldsymbol{e}_v \in \mathbb{R}^L$ are the embeddings of topic $k$ and word $v$, respectively.

---

[1]To be precise, an OT distance becomes a "distance metric" in mathematics only if the cost function $\mathbf{M}$ is induced from a distance metric. We call it "OT distance" to assist the readability of our paper.

The embeddings are expected to capture the semantic information of the topics and words. Instead of learning the word embeddings, we propose to feed them with pretrained word embeddings such as word2vec (Mikolov et al., 2013) and GloVe (Pennington et al., 2014). This not only reduces the parameter space to make the learning of $\mathbf{M}$ more stable but also enables us to leverage the rich semantic information in pretrained word embeddings, which is beneficial for short documents. Here the cosine distance instead of others is used for two reasons: it is the most commonly-used distance metric for word embeddings and the cost matrix $\mathbf{M}$ is positive thus the similarity metric requires to be upper-bounded. As cosine similarity falls in the range of $[-1, 1]$, we have $\mathbf{M} \in [0, 2]^{V \times K}$.

For easy presentation, we denote $\mathbf{G} \in \mathbb{R}^{L \times K}$ and $\mathbf{E} \in \mathbb{R}^{L \times V}$ as the collection of the embeddings of all topics and words, respectively. Now we can rewrite Eq. (4) as:

$$\min_{\theta, \mathbf{G}} d_{\mathbf{M}}(\tilde{\boldsymbol{x}}, \boldsymbol{z}) . \tag{6}$$

Although the mechanisms are totally different, both $\mathbf{M}$ of our model and $\mathbf{W}$ in NTMs (See Section 2.1) capture the relations between topics and words ($\mathbf{M}$ is distance while $\mathbf{W}$ is similarity). Here $\mathbf{M}$ is the cost function of our OT loss while $\mathbf{W}$ is the weights in the decoder of NTMs. Different from other NTMs based on VAEs, our model does not explicitly has a decoder to project $\boldsymbol{z}$ back to the word space to reconstruct $\boldsymbol{x}$, as the OT distance facilitates us to compute the distance between $\boldsymbol{z}$ and $\tilde{\boldsymbol{x}}$ directly. To further understand our model, we can actually project $\boldsymbol{z}$ to the space of $\boldsymbol{x}$ by "virtually" defining a decoder: $\phi(\boldsymbol{z}) := \text{softmax}((2 - \mathbf{M})\boldsymbol{z})$. With the notation of $\phi(\boldsymbol{z})$, we show the following theorem to reveal the relationships between other NTMs and ours, whose proof is shown in Section A of the appendix.

**Theorem 1.** *When $V \geq 8$ and $\mathbf{M} \in [0, 2]^{V \times K}$, we have:*

$$d_{\mathbf{M}}(\tilde{\boldsymbol{x}}, \boldsymbol{z}) \leq -\tilde{\boldsymbol{x}}^T \log \phi(\boldsymbol{z}). \tag{7}$$

With Theorem 1, we have:

**Lemma 1.** *Maximising the expected multinomial log-likelihood of NTMs is equivalent to minimising the upper bound of the OT distance in our model.*

Frogner et al. (2015) propose to minimise the OT distance between the predicted and true label distributions for classification tasks. It is reported in the paper that combining the OT loss with the conventional cross-entropy loss gives better performance on using either of them. As the expected multinomial log-likelihood is easier to learn and can be helpful to guide the optimisation of the OT distance, empirically inspired by Frogner et al. (2015) and theoretically motivated by Theorem 1, we propose the following joint loss for our model that combines the OT distance with the expected log-likelihood:

$$\max_{\theta, \mathbf{G}} \left( \tilde{\boldsymbol{x}}^T \log \phi(\boldsymbol{z}) - d_{\mathbf{M}}(\tilde{\boldsymbol{x}}, \boldsymbol{z}) \right) . \tag{8}$$

If we compare the above loss with the ELBO of Eq. (1), it can be observed that similar to the KL divergence of NTMs, our OT distance can be viewed as a regularisation term to the expected log-likelihood ($\tilde{\boldsymbol{x}}^T \log \phi(\boldsymbol{z}) := \frac{1}{S} \boldsymbol{x}^T \log \phi(\boldsymbol{z})$). Compared with other NTMs, our model eases the burden of developing the prior/posterior distributions and the associated sampling schemes. Moreover, with OT's ability to better modelling geometric structures, our model is able to achieve better performance in terms of both document representation and topic quality. In addition, the cost function of the OT distance provides a natural way of incorporating pretrained word embeddings, which boosts our model's performance on short documents.

Finally, we replace the OT distance with the Sinkhorn distance (Cuturi, 2013), which leads to the final loss function:

$$\max_{\theta, \mathbf{G}} \left( \epsilon \tilde{\boldsymbol{x}}^T \log \phi(\boldsymbol{z}) - d_{\mathbf{M}, \alpha}(\tilde{\boldsymbol{x}}, \boldsymbol{z}) \right) . \tag{9}$$

where $\boldsymbol{z} = \text{softmax}(\theta(\tilde{\boldsymbol{x}}))$; $\mathbf{M}$ is parameterised by $\mathbf{G}$; $\phi(\boldsymbol{z}) := \text{softmax}((2 - \mathbf{M})\boldsymbol{z})$; $\boldsymbol{x}$ and $\tilde{\boldsymbol{x}}$ are the word count vector and its normalisation, respectively; $\epsilon$ is the hyperparameter that controls the weight of the expected likelihood; $\alpha$ is the hyperparameter for the Sinkhorn distance.

To compute the Sinkhorn distance, we leverage the Sinkhorn algorithm (Cuturi, 2013). Accordingly, we name our model **N**eural **S**inkhorn **T**opic **M**odel (NSTM), whose training algorithm is shown in

**input** : Input documents, Pretrained word embeddings $\mathbf{E}$, Topic number $K$, $\epsilon$, $\alpha$
**output** : $\theta$, $\mathbf{G}$
Randomly initialise $\theta$ and $\mathbf{G}$;
**while** *Not converged* **do**
    Sample a batch of $B$ input documents $\mathbf{X}$;
    Column-wisely normalise $\mathbf{X}$ to get $\tilde{\mathbf{X}}$
    Compute $\mathbf{M}$ with $\mathbf{G}$ and $\mathbf{E}$ by Eq. (5);
    Compute $\mathbf{Z} = \mathrm{softmax}(\theta(\tilde{\mathbf{X}}))$;
    Compute the first term of Eq. (9);
    *# Sinkhorn iterations #*
    $\boldsymbol{\Psi_1} = \mathrm{ones}(K, B)/K$, $\boldsymbol{\Psi_2} = \mathrm{ones}(V, B)/V$;
    $\mathbf{H} = e^{-\mathbf{M}/\alpha}$;
    **while** $\boldsymbol{\Psi_1}$ *changes or any other relevant stopping criterion* **do**
        $\boldsymbol{\Psi_2} = \tilde{\mathbf{X}} \odot 1/(\mathbf{H}\boldsymbol{\Psi_1})$;
        $\boldsymbol{\Psi_1} = \mathbf{Z} \odot 1/(\mathbf{H}^T\boldsymbol{\Psi_2})$;
    **end**
    Compute the second term of Eq. (9): $d_{\mathbf{M},\alpha} = \mathrm{sum}(\boldsymbol{\Psi_2}^T(\mathbf{H} \odot \mathbf{M})\boldsymbol{\Psi_1})$;
    Compute the gradients of Eq. (9) in terms of $\theta$, $\mathbf{G}$;
    Update $\theta$, $\mathbf{G}$ with the gradients;
**end**

**Algorithm 1:** Training algorithm for NSTM. $\mathbf{X} \in \mathbb{N}^{V \times B}$ and $\mathbf{Z} \in \mathbb{R}_{>0}^{K \times B}$ consists of the word count vectors and topic distributions for all the documents, respectively; $\odot$ is the element-wise multiplication.

Algorithm 1. It is noteworthy that the Sinkhorn iterations can be implemented with the tensors of TensorFlow/PyTorch (Patrini et al., 2020). Therefore, the loss of Eq. (9) is differentiable in terms of $\theta$ and $\mathbf{G}$, which can be optimised jointly in one training iteration. After training the model, we can infer $z$ by conducting a forward-pass of the encoder $\theta$ with the input $\tilde{x}$. In practice, $x$ can be normalised by other methods e.g., softmax or one can use TF-IDF as the input data of the encoder.

## 4 RELATED WORKS

We first consider NTMs (e.g. in Miao et al. (2016); Srivastava & Sutton (2017); Krishnan et al. (2018); Card et al. (2018); Burkhardt & Kramer (2019); Dieng et al. (2020) reviewed in Section 2.1 as the closest line of related works to ours. For a detailed survey of NTMs, we refer to Zhao et al. (2021). Connections and comparisons between our model and NTMs have been discussed in Section 3. In addition, word embeddings have been recently widely-used as complementary metadata for topic models, especially for modelling short texts. For Bayesian probabilistic topic models, word embeddings are usually incorporated into the generative process of word counts, such as in Petterson et al. (2010); Nguyen et al. (2015); Li et al. (2016); Zhao et al. (2017). Due to the flexibility of NTMs, word embeddings can be incorporated as part of the encoder input, such as in Card et al. (2018) or they can be used in the generative process of words such as in Dieng et al. (2020). Our novelty with NSTM is that word embeddings are naturally incorporated in the cost function of the OT distance.

To our knowledge, the works that connect topic modelling with OT are still very limited. In Yurochkin et al. (2019) authors proposed to compare two documents' similarity with the OT distance between their topic distributions extracted from a pretrained LDA, but the aim is not to learn a topic model. Another recent work related to ours is Wasserstein LDA (WLDA) (Nan et al., 2019), which adapts the framework of Wasserstein AutoEncoders (WAEs) (Tolstikhin et al., 2018). The key difference from ours is that WLDA minimises the Wasserstein distance between the fake data generated with topics and real data, which can be viewed as an OT variant to VAE-NTMs. However, our NSTM directly minimises the OT distance between $z$ and $x$, where there are no explicit generative processes from topics to data. Other two related works are Distilled Wasserstein Learning (DWL) (Xu et al., 2018) and Optimal Transport LDA (OTLDA) (Huynh et al., 2020), which adapt the idea of Wasserstein barycentres and Wasserstein Dictionary Learning (Rolet et al., 2016; Schmitz et al., 2018). There are fundamental differences of ours from DWL and OTLDA in terms of the relations between

Table 1: Statistics of the datasets

|  | Number of docs | Vocabulary size (V) | Total number of words | Number of labels |
|---|---|---|---|---|
| 20NG | 18,846 | 22,636 | 2,037,671 | 20 |
| WS | 12,337 | 10,052 | 192,483 | 8 |
| TMN | 32,597 | 13,368 | 592,973 | 7 |
| Reuters | 11,367 | 8,817 | 836,397 | N/A |
| RCV2 | 804,414 | 7,282 | 60,209,009 | N/A |

documents, topics, and words. Specifically, in DWL and OTLDA, documents and topics locate in one space of words (i.e., both are distributions over words) and $x$ can be approximated with the weighted Wasserstein barycentres of all the topic-word distributions, where the weights can be interpreted as the topic proportions of the document, i.e., $z$. However, in NSTM, a document locates in both the topic space and the word space and topics and words are embedded in the embedding space. These differences lead to different views of topic modelling and different frameworks as well. Moreover, DWL mainly focuses on learning word embeddings and representations for International Classification of Diseases (ICD) codes, while NSTM aims to be a general method of topic modelling. Finally, DWL and OTLDA are not neural network models while ours is.

## 5 EXPERIMENTS

We conduct extensive experiments on several benchmark text datasets to evaluate the performance of NSTM against the state-of-the-art neural topic models.

### 5.1 EXPERIMENTAL SETTINGS

**Datasets:** Our experiments are conducted on five widely-used benchmark text datasets, varying in different sizes, including 20 News Groups (**20NG**)[2], Web Snippets (**WS**) (Phan et al., 2008), Tag My News (**TMN**) (Vitale et al., 2012)[3], **Reuters** extracted from the Reuters-21578 dataset[4], Reuters Corpus Volume 2 (**RCV2**) (Lewis et al., 2004)[5]. The statistics of the datasets in the experiments are shown in Table 1. In particular, WS and TMN are short documents; 20NG, WS, and TMN are associated with document labels[6].

**Evaluation metrics:** We report Topic Coherence (**TC**) and Topic Diversity (**TD**) as performance metrics for topic quality. TC measures the semantic coherence in the most significant words (top words) of a topic, given a reference corpus. We apply the widely-used Normalized Pointwise Mutual Information (NPMI) (Aletras & Stevenson, 2013; Lau et al., 2014) computed over the top 10 words of each topic, by the Palmetto package (Röder et al., 2015)[7]. As not all the discovered topics are interpretable (Yang et al., 2015; Zhao et al., 2018), to comprehensively evaluate the topic quality, we choose the topics with the highest NPMI and report the average score over those selected topics. We vary the proportion of the selected topics from 10% to 100%, where 10% indicates the top 10% topics with the highest NPMI are selected and 100% means all the topics are used. TD, as its name implies, measures how diverse the discovered topics are. We define topic diversity to be the percentage of unique words in the top 25 words (Dieng et al., 2020) of the selected topics, similar in TC. TD close to 0 indicates redundant topics; TD close to 1 indicates more varied topics. As doc-topic distributions can be viewed as unsupervised document representations, to evaluate the quality of such representations, we perform document clustering tasks and report the purity and Normalized Mutual Information (NMI) (Manning et al., 2008) on 20NG, WS, and TMN, where the document labels are considered. With the default training/testing splits of the datasets, we train a model on the training documents and infer the topic distributions $z$ on the testing documents. Given $z$, we

---

[2]http://qwone.com/~jason/20Newsgroups/

[3]http://acube.di.unipi.it/tmn-dataset/

[4]https://kdd.ics.uci.edu/databases/reuters21578/reuters21578.html

[5]https://trec.nist.gov/data/reuters/reuters.html

[6]We do not consider the labels of Reuters and RCV2 as there are multiple labels for one document.

[7]http://palmetto.aksw.org

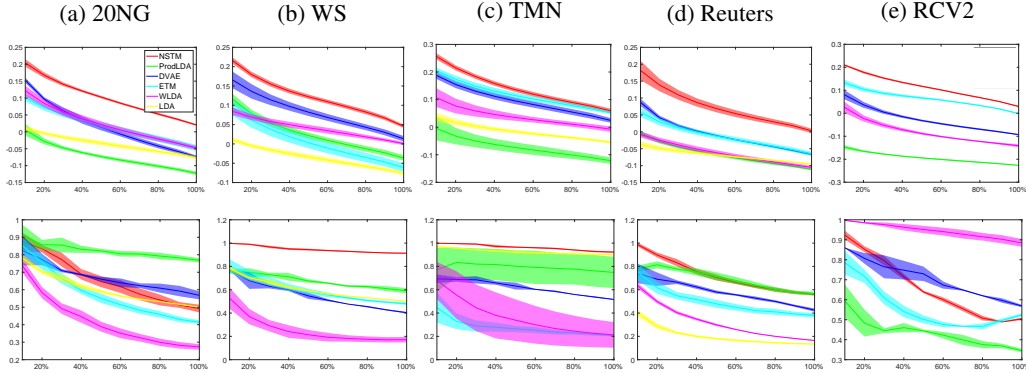

Figure 1: The first row shows the TC scores for all the datasets and the second row shows the corresponding TD scores. In each subfigure, the horizontal axis indicates the proportion of selected topics according to their NPMIs.

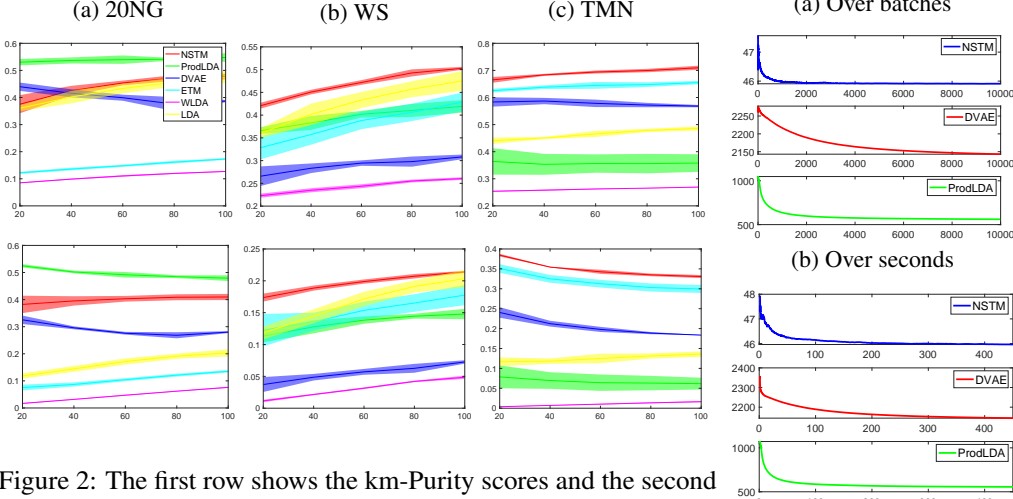

Figure 2: The first row shows the km-Purity scores and the second row shows the corresponding km-NMI scores. In each subfigure, the horizontal axis indicates the number of KMeans clusters.

Figure 3: Training loss.

adopt two strategies to perform the document clustering task: **i)** Following Nguyen et al. (2015), we use the most significant topic of a testing document as its clustering assignment to compute purity and NMI (denoted by **top-Purity** and **top-NMI**); **ii)** We apply the KMeans algorithm on $z$ (over all the topics) of the testing documents and report the purity and NMI of the KMeans clusters (denoted by **km-Purity** and **km-NMI**). For the first strategy, the number of clusters equals to the number of topics while for the second one, we vary the number of clusters of KMeans in the range of $\{20, 40, 60, 80, 100\}$. Note that our goal is not to achieve the state-of-the-art document clustering results but compare document representations of topic models. For all the metrics, higher values indicate better performance.

**Baseline methods and their settings:** We compare with the state-of-the-art NTMs, including: LDA with Products of Experts (**ProdLDA**) (Srivastava & Sutton, 2017), which replaces the mixture model in LDA with a product of experts and uses the AVI for training; Dirichlet VAE (**DVAE**) (Burkhardt & Kramer, 2019), which is a neural topic model imposing the Dirichlet prior/posterior on $z$. We use the variant of DVAE with rejection sampling VI, which is reported to perform the best; Embedding Topic Model (**ETM**) (Dieng et al., 2020), which is a topic model that incorporates word embeddings and is learned by AVI; Wasserstein LDA (**WLDA**) (Nan et al., 2019), which is a WAE-based topic model. For all the above baselines, we use their official code with the best reported settings.

**Settings for NSTM:** NSTM is implemented on TensorFlow. For the encoder $\theta$, to keep simplicity, we use a fully-connected neural network with one hidden layer of 200 units and ReLU as the activation function, followed by a dropout layer (rate=0.75) and a batch norm layer, same to the

Table 2: top-Purity and top-NMI for document clustering. The best and second scores of each dataset are highlighted in boldface and with an underline, respectively.

| | top-Purity | | | top-NMI | | |
|---|---|---|---|---|---|---|
| | 20NG | WS | TMN | 20NG | WS | TMN |
| LDA | 0.398±0.013 | 0.446±0.022 | 0.470±0.008 | 0.320±0.010 | 0.185±0.013 | 0.125±0.006 |
| ProdLDA | 0.417±0.004 | 0.293±0.023 | 0.405±0.157 | 0.321±0.004 | 0.066±0.016 | 0.091±0.101 |
| DVAE | 0.281±0.006 | 0.284±0.005 | 0.477±0.012 | 0.187±0.005 | 0.059±0.001 | 0.113±0.004 |
| ETM | 0.063±0.003 | 0.215±0.001 | 0.556±0.022 | 0.005±0.005 | 0.003±0.003 | 0.328±0.010 |
| WLDA | 0.117±0.001 | 0.239±0.003 | 0.260±0.002 | 0.060±0.001 | 0.026±0.001 | 0.009±0.001 |
| NSTM | **0.477**±0.011 | **0.451**±0.009 | **0.637**±0.010 | **0.415**±0.012 | **0.201**±0.004 | **0.334**±0.004 |

settings of Burkhardt & Kramer (2019). For the Sinkhorn algorithm, following Cuturi (2013), the maximum number of iterations is 1,000 and the stop tolerance is 0.005[8]. In all the experiments, we fix $\alpha = 20$ and $\epsilon = 0.07$. We further vary the two hyperparameters to study our model's sensitivity to them in Figure B.1 of the appendix. Finetuning the parameters specifically to a dataset may give better results. The optimisation of NSTM is done by Adam (Kingma & Ba, 2015) with learning rate 0.001 and batch size 200 for maximally 50 iterations. For NSTM and ETM, the 50-dimensional (i.e., $L = 50$, see Eq. (5)) GloVe word embeddings (Pennington et al., 2014) pre-trained on Wikipedia[9] are used. We use the number of topics $K = 100$ in most cases and set $K = 500$ on RCV2 to test our model's scalability.

## 5.2 RESULTS

**Quantitative results:** We run all the models in comparison five times with different random seeds and report the mean and standard deviation (as error bars). We show the results of TC and TD in Figure 1 and top-Purity/NMI in Table 2, and km-Purity/NMI in Figure 2, respectively. We have the following remarks about the results: **i)** Our proposed NSTM outperforms the others significantly in terms of topic coherence while obtaining high topic diversity on all the datasets. Although others may have higher TD than ours in one dataset or two, they usually cannot achieve a high TC at the same time. **ii)** In terms of document clustering, our model performs the best in general with a significant gap over other NTMs, except the case where ours is the second for the KMeans clustering on 20NG. This demonstrates that NSTM is not only able to discover interpretable topics with better quality but also learn good document representations for clustering. It also shows that with the OT distance, our model can achieve a better balance among the comprehensive metrics of topic modelling. **iii)** For all the evaluation metrics, our model is consistently the best on the short documents including WS and TMN. This demonstrates the effectiveness of our way of incorporating pretrained word embeddings, which shows our model's potential on short text topic modelling. Although ETM also uses pretrained word embeddings, its performance is incomparable to ours.

**Scalability:** NSTM has comparable scalability with other NTMs and is able to scale on large datasets with a large number of topics. To demonstrate the scalability, we run NSTM, DVAE, ProdLDA (as these three are implemented in TensorFlow, while ETM is in PyTorch, and WLDA is in MXNet) on RCV2 with $K = 500$. The three models run on a Titan RTX GPU with batch size 1,000. Figure 3 shows the training losses, which demonstrate that NSTM has similar learning speed to ProdLDA, better than DVAE. The TC and TD scores of this experiment are shown in Section C of the appendix, where it can be observed that with 500 topics, our model shows similar performance advantage over others.

**Qualitative analysis:** As topics in our model are embedded in the same space as pretrained word embeddings, they share similar geometric properties. Figure 4 shows a qualitative analysis. For the t-SNE (Maaten & Hinton, 2008) visualisation, we select the top 50 topics with the highest NPMI learned by a run of NSTM on RCV2 with $K = 100$ and feed their (50 dimensional) embeddings into the t-SNE method. We also show the top five words and the topic number (1 to 50) of each topic. We

---

[8]The Sinkhorn algorithm usually reaches the stop tolerance in less than 50 iterations in NSTM

[9]https://nlp.stanford.edu/projects/glove/

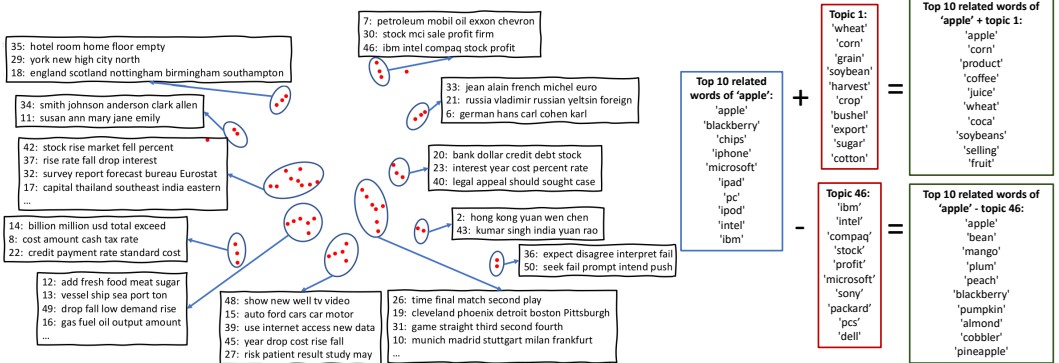

Figure 4: Left: t-SNE visualisation of topic embeddings on RCV2. One red dot represents a topic. The top 5 words and the topic number (1 to 50) of each topic are also shown. Right: interactions between word and topic embeddings.

can observe that although the words of the topics are different, the semantic similarity between the topics captured by the embeddings is highly interpretable. In addition, we take the GloVe embeddings of the polysemantic word "apple" and find the closest 10 related words among the 0.4 million words of the GloVe vocabulary according to their cosine similarity. It can be seen that by default "apple" refers to the Apple company more in GloVe. Either adding the embeddings of topic 1 that describes the concept of "food" or subtracting the embeddings of topic 46 that describes the concept of "tech companies" reveals the fruit semantic for the word "apple". More qualitative analysis on topics are provided in Section E of the appendix.

## 6 CONCLUSION

In this paper, we presented a novel neural topic model based on optimal transport, where a document is endowed with two representations: the word distribution, $\tilde{x}$, and the topic distribution, $z$. An OT distance is leveraged to compare the semantic distance between the two distributions, whose cost function is defined according to the cosine similarities between topics and words in the embedding space. $z$ is obtained from an encoder that takes $\tilde{x}$ as input and is trained by minimising the OT distance between $z$ and $\tilde{x}$. With pretrained word embeddings, topic embeddings are learned by the same minimisation of the OT distance in terms of the cost function. Our model has shown appealing properties that are able to overcome several shortcomings of existing neural topic models. extensive experiments have been conducted, showing that our model achieves state-of-the-art performance on both discovering quality topics and deriving useful document representations for both regular and short texts.

### ACKNOWLEDGMENTS

Trung Le was supported by AOARD grant FA2386-19-1-4040. Wray Buntine was supported by the Australian Research Council under award DP190100017.

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

## A    PROOF OF THEOREM 1

*Proof.* Before showing the proof, we introduce the following notations: We denote $k \in \{1, \cdots, K\}$ and $v \in \{1, \cdots, V\}$ as the indexes; The $s^{\text{th}}$ ($s \in \{1, \cdots, S\}$) token of the document picks a word in the vocabulary, denoted by $w_s \in \{1, \cdots, V\}$; the normaliser in the softmax function of $\phi(\boldsymbol{z})$ is denoted as $\hat{\phi}$ so:

$$\hat{\phi} = \sum_{v=1}^{V} e^{\sum_{k=1}^{K} z_k(2-m_{vk})} = e^2 \sum_{v=1}^{V} e^{-\sum_{k=1}^{K} z_k m_{vk}} .$$

With these notations, we first have the following equation for the multinomial log-likelihood:

$$\tilde{\boldsymbol{x}}^T \log \phi(\boldsymbol{z}) = \frac{1}{S} \sum_{s=1}^{S} \log \phi(\boldsymbol{z})_{w_s}$$

$$= \frac{1}{S} \sum_{s=1}^{S} \left( \sum_{k=1}^{K} z_k(2 - m_{w_s k}) - \log \hat{\phi} \right)$$

$$= 2 - \log \hat{\phi} - \frac{1}{S} \sum_{s=1}^{S} \sum_{k=1}^{K} z_k m_{w_s k} . \qquad (A.1)$$

Recall that in Eq. (1) of the main paper, the transport matrix $\mathbf{P}$ is one of the joint distributions of $\tilde{\boldsymbol{x}}$ and $\boldsymbol{z}$. We introduce the conditional distribution of $\boldsymbol{z}$ given $\tilde{\boldsymbol{x}}$ as $\mathbf{Q}$, where $q(v, k)$ indicates the probability of assigning a token of word $v$ to topic $k$.

Given that $\mathbf{P}$ satisfies $\mathbf{P} \in U(\tilde{\boldsymbol{x}}, \boldsymbol{z})$ and $p_{vk} = \tilde{x}_v q(v, k)$, $\mathbf{Q}$ must satisfy $U'(\tilde{\boldsymbol{x}}, \boldsymbol{z}) := \{\mathbf{Q} \in \mathbb{R}_{>0}^{V \times K} | \sum_{v=1}^{V} \tilde{x}_v q(v, k) = z_k\}$. With $\mathbf{Q}$, we can rewrite the OT distance as:

$$d_{\mathbf{M}}(\tilde{\boldsymbol{x}}, \boldsymbol{z}) = \min_{\mathbf{Q} \in U'(\tilde{\boldsymbol{x}}, \boldsymbol{z})} \sum_{v=1, k=1}^{V, K} \tilde{x}_v q(v, k) m_{vk}$$

$$= \frac{1}{S} \min_{\mathbf{Q} \in U'(\tilde{\boldsymbol{x}}, \boldsymbol{z})} \sum_{k=1}^{K} \sum_{s=1}^{S} q(w_s, k) m_{w_s k}.$$

If we let $q(v, k) = z_k$, meaning that all the tokens of a document to the topics according to the document's doc-topic distribution, then $\mathbf{Q}$ satisfies $U'(\tilde{\boldsymbol{x}}, \boldsymbol{z})$, which leads to:

$$d_{\mathbf{M}}(\tilde{\boldsymbol{z}}, \boldsymbol{x}) \le \frac{1}{S} \sum_{k=1}^{K} \sum_{s=1}^{S} z_k m_{w_s k} . \qquad (A.2)$$

Together with Eq. (A.1), the definition of $\hat{\phi}$, and the fact that $m_{vk} \le 2$, we have:

$$\tilde{\boldsymbol{x}}^T \log \phi(\boldsymbol{z}) = 2 - \log \hat{\phi} - \frac{1}{S} \sum_{s=1}^{S} \sum_{k=1}^{K} z_k m_{w_s k}$$

$$\le -\log \left( \sum_{v=1}^{V} e^{-\sum_{k=1}^{K} z_k m_{vk}} \right) - d_{\mathbf{M}}(\tilde{\boldsymbol{x}}, \boldsymbol{z})$$

$$\le -(\log V - 2) - d_{\mathbf{M}}(\tilde{\boldsymbol{x}}, \boldsymbol{z})$$

$$\le -d_{\mathbf{M}}(\tilde{\boldsymbol{x}}, \boldsymbol{z}) , \qquad (A.3)$$

where the last equation holds if $\log V > 2$, i.e., $V \ge 8$. □

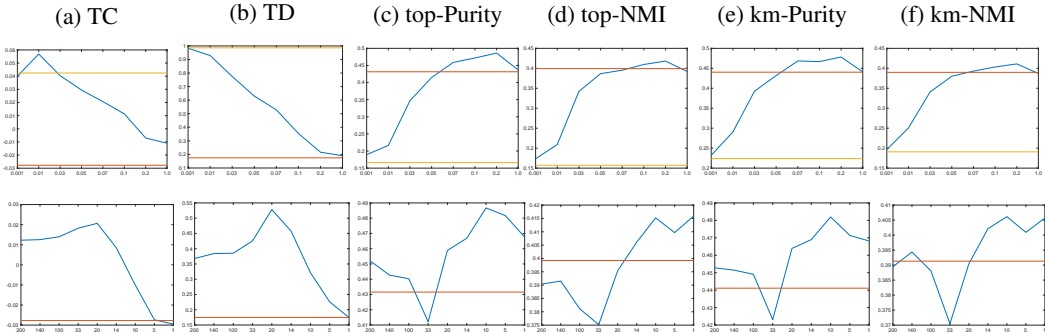

Figure B.1: Parameter sensitivity of NSTM on 20News. The first and second show the performance with different values of $\epsilon$ and $\alpha$, respectively. In the first row, we fix $\alpha = 20$ and vary $\epsilon$ while in the second row, we fix $\epsilon = 0.07$ and vary $\alpha$.

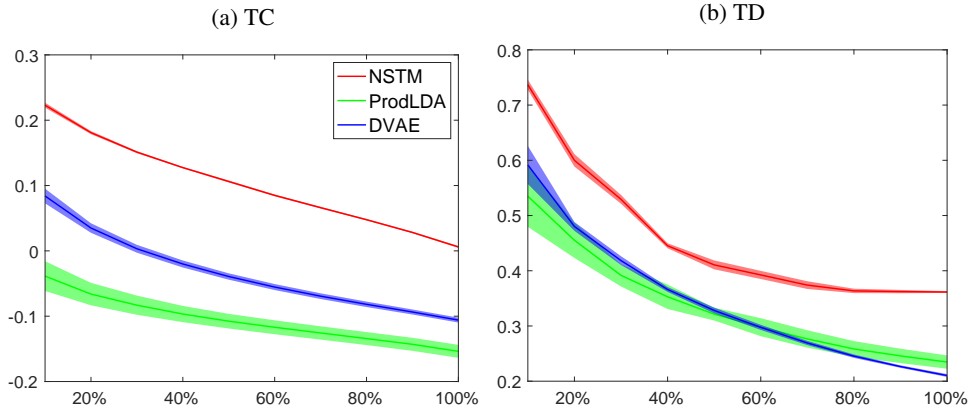

Figure C.1: TC and TD on RCV2 with 500 topics.

## B  PARAMETER SENSITIVITY

In the previous experiments, we fix the values of $\epsilon$ and $\alpha$, which control the weight of the multinomial likelihood in Eq (9) and the weight of the entropic regularisation in the Sinkhorn distance, respectively. Here we report the performance of NSTM on 20NG (blue lines) under different settings of the two hyperparameters in Figure B.1. Moreover, we propose two variants of NSTM. The first one removes the Sinkhorn distance in the training loss of Eq. (9) (i.e., only the expected log-likelihood term left) and its performance is shown as the red lines. The second variant removes the the expected log-likelihood term in the training loss of Eq. (9) (i.e., only Sinkhorn distance left) and its performance is shown as yellow lines.

## C  TC AND TD ON RCV2 WITH 500 TOPICS

The results are shown in Figure C.1.

## D  AVERAGE SINKHORN DISTANCE WITH VARIED NUMBER OF TOPICS

In Figure D.1, we show the average Sinkhorn distance with varied number of topics on 20NG, WS, TMN, and Reuters. It can be observed that when $K$ increases, there is a clear trend that $d_{\mathbf{M}}(\boldsymbol{z}, \boldsymbol{x})$ decreases.

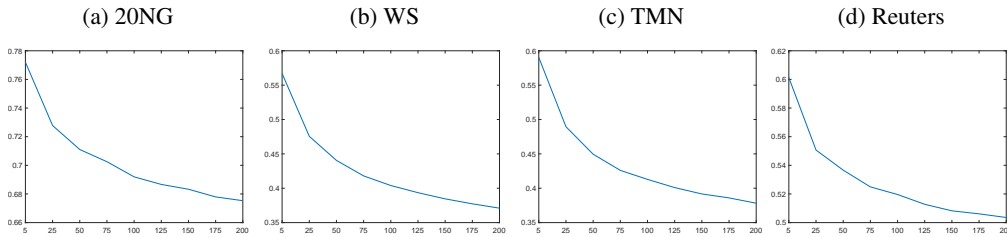

Figure D.1: Sinkhorn distance with varied $K$. Vertical axis: the average Sinkhorn distance over all the training documents, i.e., mean $d_{\mathbf{M}}(\boldsymbol{z}, \boldsymbol{x})$. Horizontal axis: the number of topics, i.e., $K$
.

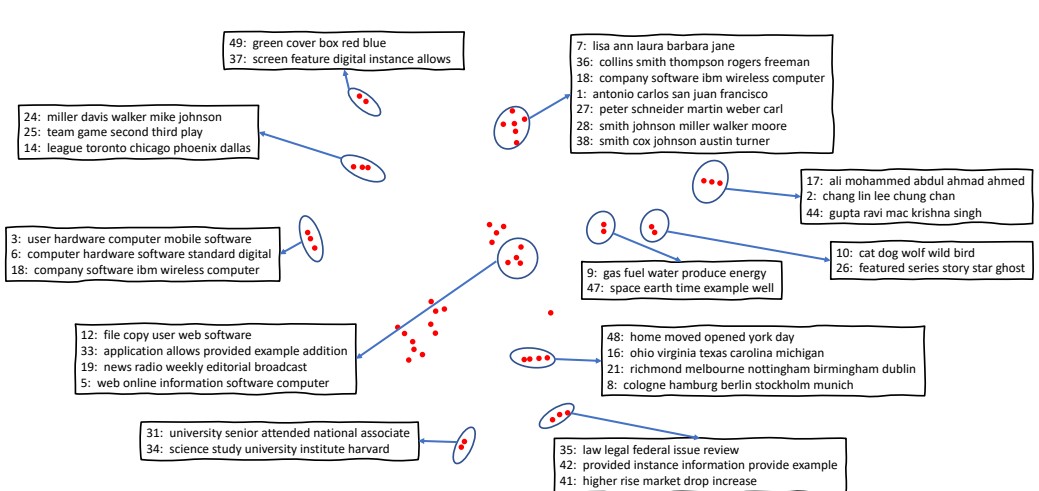

Figure E.1: t-SNE visualisation of topic embeddings on 20NG.

## E   MORE TOPIC EMBEDDING VISUALISATIONS

In Figure E.1, E.2, E.3, and E.4, we show the visualisations of 20NG, WS, TMN, and Reuters, respectively. We note that the topic embeddings in general present much better clustering structures of topics in the semantic space. Such topic correlations can only be detected by specialised topic models (e.g.,in Lafferty & Blei (2006); Blei et al. (2010); Zhou et al. (2016)). Instead, the correlations of topics in our model are implicitly captured by the semantic embeddings.

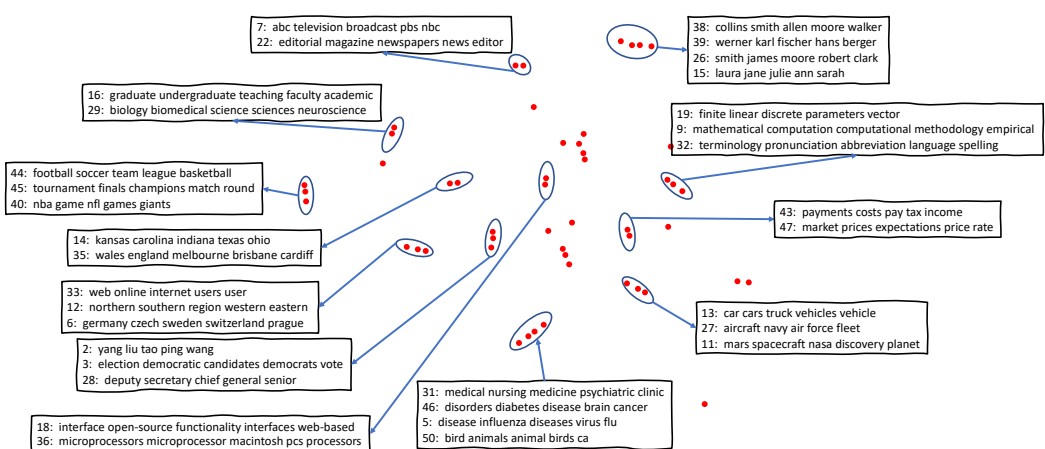

Figure E.2: t-SNE visualisation of topic embeddings on WS.

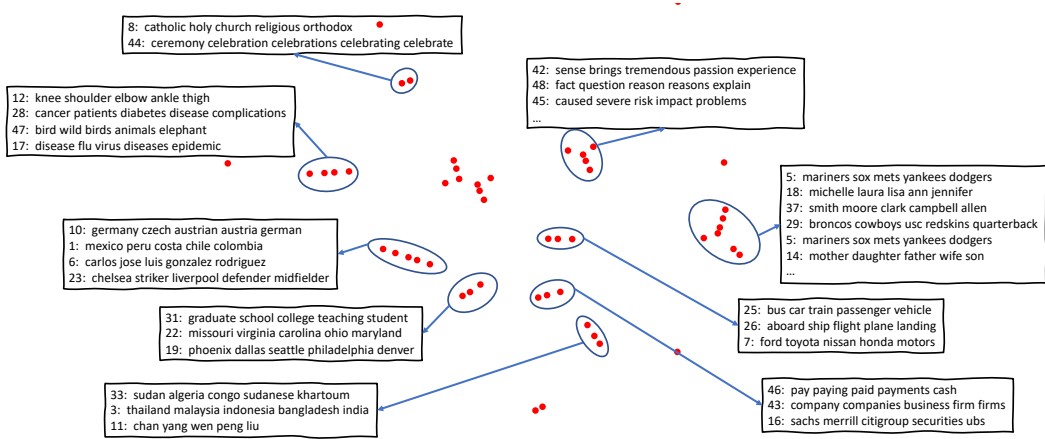

Figure E.3: t-SNE visualisation of topic embeddings on TMN.

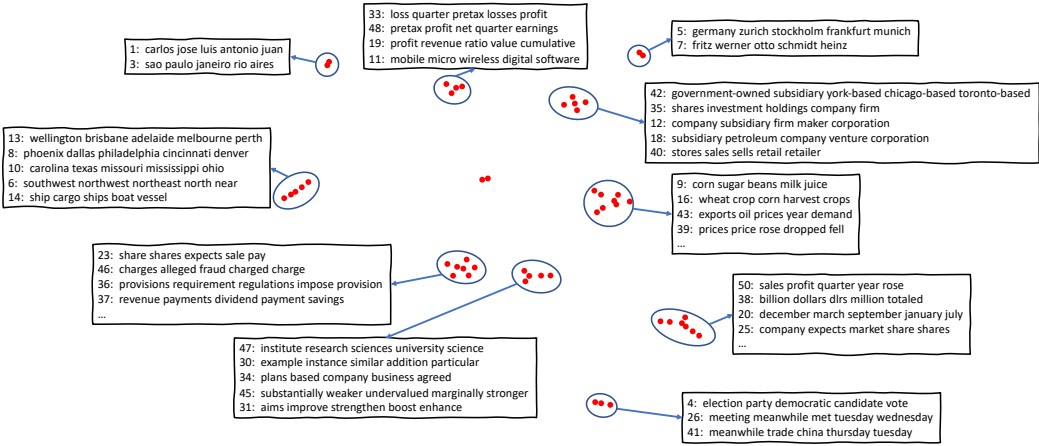

Figure E.4: t-SNE visualisation of topic embeddings on Reuters.

