# OpenReview forum: "Neural Topic Model via Optimal Transport"
_ICLR.cc/2021/Conference — ICLR 2021 Spotlight_

### Official Review · AnonReviewer2 · 2020-10-26
**Impressive empirical results, but discussion of optimal transport lacks depth**

**Rating:** 7
**Confidence:** 3

**Review:**

Summary: The paper proposes a neural topic model which log-likelihood is regularized by Sinkhorn distance, instead of following Variational AutoEncoder (VAE) approach. The proposed model is hence cannot be interpreted as a probabilistic generative model. Still, with respect to metrics such as Topic Coherence and Topic Diversity which don't require probabilistic interpretation of topic model, the proposed model performs very well across five standard benchmark datasets for topic modeling.

Quality and Clarity: At the very high level, the use of Optimal Transport distance for topic modeling is sensible. Neither word counts nor topic distributions have clear ordering of indices, and Optimal Transport distance is usually well-suited for such histogram-like features. However, authors do not provide much discussion on why Optimal Transport distance is particularly well-suited for topic modeling. I can see that compared to the expected log-likelihood, the Optimal Transport distance $d_M(\tilde{x},z)$ is a more optimistic estimate, as the topic distribution for each word could be individually assigned as long as marginal distribution of topics is equal to $z$. Maybe this is related to some sort of variational approximation which disentangles the usual constraint that topic distribution of all words in the same document are exactly the same. Still, I had hard time understanding why this is favorable for topic modeling, or finding better interpretations of Optimal Transport distance on topic modeling. At the least, some qualitative analysis of how the hyperparameter $\varepsilon$, which controls the tradeoff between expected likelihood and Sinkhorn distance, affects the characteristics of topic models would be nice. Appendix D. shows how evaluation metrics change as $\varepsilon$ changes, but not much interpretation of results is provided.

It's good to see how a well-established technique can be applied to topic modeling and show encouraging empirical results, but the lack of justifications on the modeling approach would make it difficult for the further development and adoption of the method, especially because the proposed method is not a probabilistic generative model anymore. I would encourage others to do a deeper qualitative analysis on the effect of Sinkhorn distance as a regularizer.

Originality and Significance: The establishment of the connection between Optimal Transport and Topic Modeling would potentially lead to active follow-up research, as both are well-established areas of research.

* Pros:
  * good empirical results
  * connects two well-established topics of research
* Cons:
  * Justifications of the modeling approach is lacking

---

### Official Review · AnonReviewer1 · 2020-10-27
**a topic model with compelling experimental results especially for short text documents**

**Rating:** 7
**Confidence:** 3

**Review:**

This paper builds off existing neural topic model work that's based on variational autoencoders and specifically introduces a novel loss function based on optimal transport (more specifically, the Sinkhorn distance). The paper demonstrates impressive experimental results, where the proposed method Neural Sinkhorn Topic Model (NSTM) outperforms (or is competitive with) a number of SOTA baselines, with a dramatic performance gain in the setting where the text documents are very short.

Overall, I enjoyed reading this paper and found the proposed method well-described albeit somewhat ad hoc, although the authors do justify certain aspects of the method (e.g., Lemma 1). The "weight" of this paper seems to really be carried by the experimental results though, which I found to be quite compelling.

Strengths:
- well-explained method that provides a neat usage of optimal transport in topic models
- Theorem 1/Lemma 1 provide a nice theoretical comparison of the proposed method vs other neural topic models
- strong experimental performance especially in the setting of short documents, which is very relevant in practice

Weaknesses:
- the proposed method comes off as a bit ad hoc
- while the authors do reference Card et al's paper, I think a comparison with Card et al's Scholar framework would be helpful to get a sense of how the l1 regularization in Scholar/SAGE handles short text documents compared to NSTM
- there are small typos/grammar glitches here and there - please proofread carefully

In the conclusion, the authors point out correlated topic models and dynamic topic models as variants worth exploring next. Supervised versions of topic models I think also are worth exploring.

---

### Official Review · AnonReviewer4 · 2020-10-31
**Interesting alternative approach to neural topic model based on optimal transport with good demonstrated performance**

**Rating:** 8
**Confidence:** 4

**Review:**

The paper proposes a neural topic model derived from the perspective of optimal transport (OT). Topic embeddings are learned as part of the training process and is used to construct the cost matrix of the transport.  The cost function based on the OT distance is further improved by combining with the cross-entropy loss and by using the Sinkhorn distance to replace the OT distance.

The paper is well written.  The proposed method is first explained from the perspective from the optimal transport and then developed by considering the cross-entropy loss and the Sinkhorn distance.  It also explains how the model incorporates the word embedding and introduce the topic embedding to simply the cost matrix M of the transport.  The proposed method is sound.

The experiments included recent neural topic models for comparison.  The chosen test data sets include also some with short text. In general, the proposed method has been show to perform better than other methods in terms of topic coherence and topic diversity.  The experiment section also some quality analysis on the topic discovered by the proposed model.  The experimental results are convincing.

The paper explains quite clearly the relationship between the proposed method and related methods. In particular, the paper seems to have adequately credited the sources of ideas during the development of the proposed model.  The novelty of the proposed method appears to be the use of optimal transport for developing neural topic model and the construction of the cost matrix M based on the cosine similarity between word embedding and topic embedding.  The novelty is sufficient.

Although the clarity of the proposed method is good, the rationale for using OT distances for comparing probabilities may deserve more explanation.  The paper gives an intuition for the transport matrix P in section 2.2.  It may be better if it can also give more explanation on the role of M and what the meaning of the distance between two probability vector is.  References may also be given to other works that use OT distances for comparing probabilities.

The experiment includes recent neural topic models but does not include other recent topic models not based on neural networks. It would be interesting to see how the proposed method performs compared to those models.

I cannot find any indication that source code will be released.  It is suggested to do so for reproducibility and for the use of practitioners.

Minor comment:

- Add the data set name to the caption of Figure 4.

---

### Official Review · AnonReviewer3 · 2020-11-01
**Generally interesting, but comparison is not persuasive enough**

**Rating:** 6
**Confidence:** 4

**Review:**

This paper proposes a new variant of neural topic model leveraging optimal
transport in order to incorporate information from pre-trained word vectors.
Specifically, the authors replaced the KL-divergence reguralization term with
an optimal transport between topic distribution and empirical word distribution
in each text.
Experimental evaluation yields generally better topic coherence and high
precision of K-means clustering on the induced topic distributions.

Basically this paper is interesting, but still leaves some questions.

- First of all, evaluations are based only on high NPMI topics (section 5.2).
Therefore, it is trivial that the induced topic-word distribution of these
topics are good. In my experience, original vanilla LDA yields mostly
interpretable topic-word distributions; however, even for the high-NPMI topics,
each topic in Figure 4 is somewhat noisy, and unshown topics might be worse.
Therefore, I would like to know the comparison between the proposed method and
the original LDA too.

- The paper first says that "good document representation and coherent/diverse
topics" is difficult. Then why not including perplexity evaluation in the
experiments? K-means results are only auxiliary evaluation of the former, thus
the reader would like to know whether the proposed model could yield better
perplexity on documents or not.

- It seems that the choice of dimensionality and the number of topics seems
too low and arbitrary. Why only the 50-dimensional word vectors are used?
Is there any difference over the competitors when that dimensionality is
changed?
Also, I could not know why only the results with K=100 is shown in main text.
Appendix E shows that K=500 consistently yields better results; why are they
not included?

- Finally, I cannot understand what kind of optimization w.r.t M is conducted
in this paper. To make the paper as self-contained as possible, I strongly
recommend to show what kind of optimization is actually done.

The proposed OT regularization seem to work better, but I cannot see why the
baseline of word-vector based topic models like (Dieng+ 2020) is inferior.
Is there any intuition or explanation over these trivial competitors?

---

### Comment · ~marco_cuturi2 · 2021-04-12
**Congrats on the accept, suggestion of reference**

Hello,
Congrats on this accept! this is a nice paper, and it's very exciting to see OT used again in an NLP application :)

I just wanted to suggest a reference by a former student of mine, that had worked on something similar (though simpler, because no NN involved), i.e. learning topics using a Wasserstein loss. I hope this is useful.

http://proceedings.mlr.press/v51/rolet16.html

It's conceptually simpler than some of the other reference you kindly cited (thinking in particular of Schmitz et al. 2018 which also considers W combinations) but because it is, AFAIK, the first reference blending OT and text topic models, I think this would be a relevant addition to your bibliography.

Best wishes
Marco

---

> ### Comment · ~He_Zhao1 · 2021-04-13
> **Thanks for the reference**
>
> Hi Prof Cuturi,
>
> Thanks a lot for the suggestion of the reference, which we will add to our bibliography.
>
> Best regards,
> He Zhao

---

### Comment · ~He_Zhao1 · 2021-04-15
**Code of the paper**

The code of our paper is released at https://github.com/ethanhezhao/NeuralSinkhornTopicModel

---

### Comment · ~He_Zhao1 · 2022-06-01
**Correction of LDA's results on 20NG**

In the ICLR version of the paper, we plotted the wrong curves for LDA on 20NG (in the first column of Figures 1 and 2) while the other datasets were correct. The corrections have been updated in the Arxiv version (https://arxiv.org/pdf/2008.13537.pdf), where LDA has higher clustering results on 20NG.

---

### Decision · Program_Chairs · 2021-01-07
**Final Decision**

**Decision:**

Accept (Spotlight)

**Comment:**

The reviewers unanimously agreed that this is an interesting paper that belongs at ICLR. The use of optimal transport in neural topic models is novel and the paper is well-written.

A common theme among the reviewers was that they would like to see more intuition and justification. I suggest you bear this in mind while editing the final version of the paper. I also believe that R3 brings up valid points about evaluating perplexity -- I don't think the lack of perplexity results are a reason to reject the paper, but I believe they can be calculated here (see eg the reference R3 provided) and they would give a clearer view of the model's performance.